# Mangrove species classification using a proposed ensemble U-Net model and Planet satellite imagery: A case study in Ngoc Hien district, Ca Mau province, Vietnam

Tran Dang Hung[1], Minh Hai Pham[2]*, Bui Thanh Huyen[1], Tran Hong Hanh[3], Pham Hong Tinh[4], Nguyen Thanh Bang[1], Tran Thanh Tung[5]

1 Vietnam Institute of Meteorology, Hydrology and Climate Change, Vietnam Ministry of Agriculture and Environment, Ha Noi, Vietnam, 2 National Remote Sensing Department, Vietnam Ministry of Agriculture and Environment, Ha Noi, Vietnam, 3 Ha Noi University of Mining and Geology, Ha Noi, Vietnam, 4 Faculty of Environment, Hanoi University of Natural Resources and Environment, Ha Noi, Vietnam, 5 Thuy Loi University, Ha Noi, Vietnam

* phamminhhai.vigac@gmail.com

## Abstract

Land cover and plant species identification using satellite images and deep learning approaches have recently been a widely addressed area of research. However, mangroves, a specific species that have significantly declined in quantity and quality worldwide despite their numerous benefits, have not been the subject of attention. The novelty of this research is to deal with this species based on an advanced deep learning solution (a proposed ensemble U-Net model) and a high-resolution Planet satellite imagery (5 m x 5 m) in a case study of Ngoc Hien district, Ca Mau province, Vietnam. Twelve single U-Net backbone models were trained, and three quantitative metrics (Intersection over Union, $F_1$-score, and Overall Accuracy) were used to evaluate. The findings indicate that three out of twelve models (MobileNet, SEResNeXt-101 and Efficientnet-B7) experienced the most efficient assessment results for identifying all classes, in which the MobileNet model was the best. These models were applied for the ensemble model's development. The ensemble model's quantitative assessment metrics increased considerably by about 3–10% compared to the single-component models. The IoU, $F_1$-score, and OA values of this model were 80.08%, 95.82%, and 95.90%, respectively. Three classes of mangrove species (*Avicennia alba*, *Rhizophora apiculate,* and mixed mangroves) in the ensemble model had more uniform assessment results. In conclusion, to achieve optimal classification outcomes, a land-cover map comprising mangrove species is possibly established using the proposed ensemble model, while a distribution map of mangrove species enables to be developed using the MobileNet model.

**Data availability statement:** All data are in the manuscript and Supporting Information files.

**Funding:** Our study is supported by the Vietnam Ministry of Agriculture and Environment (project TNMT.ĐL.2023.04) received by Dr. Pham Hong Tinh.

**Competing interests:** The authors have declared that no competing interests exist.

## Introduction

In Vietnam, about 160,000 ha of mangrove forest, distributed along the coastline of 29 provinces and cities from Quang Ninh to Kien Giang, brings many significant benefits [1–3]. Nevertheless, the impacts of war (such as the Second Indochina War), population explosion and economic development (such as the shrimp aquaculture boom), along with climate change and extreme weather events, have resulted in dramatic variations in both the quality and quantity of mangrove forests [4]. The extent of mangrove forests in some regions considerably decreased, particularly in two communes (Tan An Tay and Tam Giang Tay) inside Ngoc Hien district of Ca Mau province. The mangrove area in these communes witnessed degradation of 490.2 ha and 603.6 ha between 2015 and 2020, respectively [5]. Mangrove degradation raises the threat to the safety of people residing in coastal areas through coastal erosion, floods, storms, and saltwater intrusion. At the same time, it affects the increase in $CO_2$ concentration, accelerating the global warming process [6]. As a result, it is imperative to analyze and provide prompt data on land cover and mangrove distribution for natural resources and environmental research and management.

Up to this point, geospatial analysis has been considered an effective approach for processing, analyzing, and providing data for mangrove studies and management. Based on remote sensing and GIS (Geographical Information System) technologies, optical characteristics of satellite imagery, and conventional machine learning algorithms (RF – Random Forest, DT – Decision Tree, or SVM – Support Vector Machine), various research and governmental projects have been implemented on the mangrove forest in Vietnam. In 2022, the study of Hai et al. [3] applying SPOT images and RF algorithms to classify mangrove species and monitor mangrove health in Ca Mau province, or the research of Tinh et al. [4] analyzing high-resolution WorldView-2 images to quantify changes in the mangrove forest of the Mekong Delta from 2015 to 2020 were notable examples. They proved that remote sensing and machine-learning methods supply data on the historical and present state of mangrove forests, effectively monitoring and detecting changes in mangrove forests [4,7–11]. However, these approaches, which entail surveying, calculating land areas, manually sampling, and generating thematic maps and reports, are still constrained by time, expenses, and biases. Notably, in the classification process, without attention to the image's structure and color, errors arise in classifying objects with similar spectra, such as distinguishing between water surfaces and aquaculture areas, newly planted forests and agricultural areas, as well as bare land and harvested agricultural areas. To address these problems, deep learning, including the U-Net model and satellite imagery with higher spatial resolution and more detailed components (such as Sentinel [12] or UAV [13]), is initially being researched.

Deep learning (DL) is a subset of machine learning (ML) that focuses on simulating the intricate decision-making abilities of the human brain. Multiple layers of interconnected nodes constitute deep neural networks, and each layer builds on the previous one before it to improve and optimize the classification or prediction [14,15]. Various categories of deep neural networks exist to tackle particular issues or datasets, in which Convolutional Neural Networks (CNNs) are primarily utilized

in computer vision with image classification or object detection tasks, including land-use/ land-cover classification. CNNs have the capability to handle complex and multi-dimensional data, automatically detect significant contextual features, and transfer data across layers, resulting in more effective data processing. Although the pooling layer leads to information loss, the advantages of using CNNs, such as reducing complexity, improving efficiency, and minimizing the danger of over-fitting, outweigh this drawback [14]. Recent literature reveals several successful attempts to apply DL-based land-use identification using satellite images. Harbas et al. (2018) [16] applied Fully Convolutional Networks (FCN) to detect roadside vegetation in RGB color images without using special equipment. Liu et al. (2018) [17] experimented with implementing deep learning models (FCN and DCNN – patch-based Deep Convolutional Neural Networks) and traditional supervised learning models (RF and SVM) to classify seven natural land-cover types. The results showed that compared to the conventional machine learning models, DCNN and FCN performed better when the sample size was large or similar, respectively. In the agriculture sector, Barbosa et al. (2020) [18] developed a CNN model to capture the spatial structure of farm attributes and model the response to nutrient and seed rate management through the growing season.

Besides, U-Net is a specialized form of CNN designed for image segmentation tasks. It enhances the capacity of image classification by allowing accurate predictions at the pixel level. It performs this with a unique design that consists of an encoder-decoder network with skip connections. Moreover, U-Net models often leverage backbones, pre-trained CNNs, to enhance their performance, integrating them into the encoder path to capture rich hierarchical features. This relationship is outstanding because the pre-trained backbones, such as ResNet or EfficientNet, provide robust and well-generalized features, improving the segmentation accuracy and training efficiency of U-Net models. The combination of U-Net's architecture and robust backbones enables accurate and intricate segmentation, making it very efficient for applications such as medical analysis and land-cover categorization. U-Net's ability to accurately segment, classify and label different land-cover classes from high-resolution remote sensing images makes it stand out. Various research studies have reported the excellent performance of U-Net backbone models, such as rice disease detection [19], land-use classification [13], or urban classification [20]. Particularly in the agriculture sector, Mahakalanda et al. (2022) [21] successfully applied DL techniques (FCNs, VGG-16 – Visual Geometry Group from Oxford, and U-Net) and remote sensing images (Setinel-2A and Sentinel-2B) to determine the stand age of rubber plantations in Sri Lanka. Shah et al. (2023) [19] detected rice disease early by comparing Inception-V3, VGG-16, VGG-19, CNN and ResNet-50. On the other hand, each U-Net backbone has a different network architecture, advantages, and disadvantages. A U-Net backbone model enables it to perform well in some classes and poorly in others. An ensemble model merging various U-Net backbone models is a powerful solution to reduce high variance and bias and improve predictions [22,23]. However, this approach has not been widely used in land-cover classification and has never been particularly applied to mangrove forests.

As a result, the main objectives of this study were: (1) to apply the superiority of deep learning for land cover comprising mangrove species classification over traditional classification methods, (2) to enhance classification performance and prediction efficiency of deep learning approach by a proposed ensemble U-Net model combining multiple single U-Net backbone models. The study was experimentally conducted in Ngoc Hien district, Ca Mau province, Vietnam, with three main types of mangrove forests (*Avicennia alba*, *Rhizophora apiculata*, and mixed mangroves). In further future, the findings are expected to be widely applied throughout Vietnam and worldwide, assisting managers and ecological planners by providing precise and timely data, improving the efficiency of land-cover monitoring, and preserving the long-term sustainability of mangrove forests.

## Materials and methods

The study possessed three primary stages: (1) Data pre-processing, (2) Deep learning model training, and (3) Proposed ensemble model and evaluation. Initially, the input dataset for stage (2) was generated by collecting and pre-processing the original and labelled images. The original image was created from the Planet satellite imagery with a resolution of 5 m x 5 m. The labelled image was developed based on our classified land-cover map that was conducted using data from Planet imagery and ground truth. In stage (2), twelve U-Net backbone models were trained after patchifying the input

datasets and dividing them into training and validation sets with a 75:25 ratio. Twelve backbones were chosen as pre-trained steps for the models based on the number of parameters and depths of various kinds of backbones on Google Colab. Overall accuracy (OA) was assessed and compared between the trained models. Finally, stage (3) encompassed the construction of a proposed ensemble model to map land cover, particularly mangrove species, by integrating three trained U-Net backbone models with strong OA values. The generalization ability of the proposed ensemble model and the trained single-component models was assessed quantitatively (with metrics: the intersection of union (IoU), $F_1$-score, and OA) and visually. The technical flowchart of this study is shown in Fig 1.

## Dataset

This study was carried out in Ngoc Hien district, Ca Mau province, Vietnam, which owns an outstanding 50,848 ha of mangrove forests, with the dominant species being *Avicennia alba* and *Rhizophora apiculate* [4,24,25]. The dataset included two images: an original image (a Planet satellite image) and a labelled image (a classified land-cover image) (Fig 2). The original image, with an average cloud cover of less than 5%, was downloaded from Planet Labs PBC (https://www.planet.com/base-maps/). The labelled image was classified with a Kappa of 88.4% using Planet satellite imagery and ground truth data. Six classified classes included *Avicennia alba*, *Rhizophora apiculata*, mixed mangroves, aquaculture, buildings, and sea/river. The ground truth data was collected in June 2022, corresponding to the period the original image was downloaded, and described land cover at 100 randomly selected reference points that contained 70 points of mangrove forest. Each image exhibited a resolution of 5 meters and a size of 12,628 x 5514. After augmentation, 550 tiles (256x256) of the original image and 550 tiles (256x256) of the labelled image were created. Then, a 75:25 ratio was used to divide them randomly into training and validation datasets, in which the mangrove layers were encountered in 120 and 50 images, respectively.

## Deep learning segmentation using a U-Net model

U-Net [26–29], a symmetrical U-shaped structure, is intended for image classification and segmentation. The contracting path reduces spatial dimensions and successfully captures context by employing a standard convolutional network

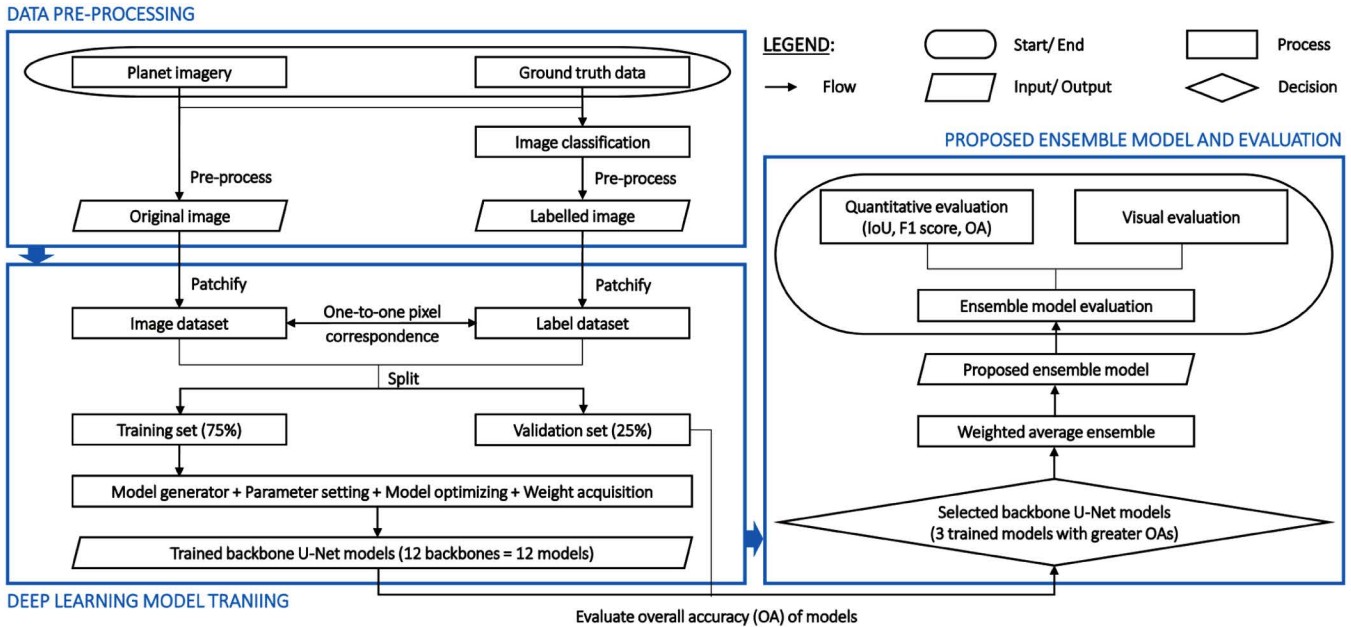

**Fig 1. Research framework.**

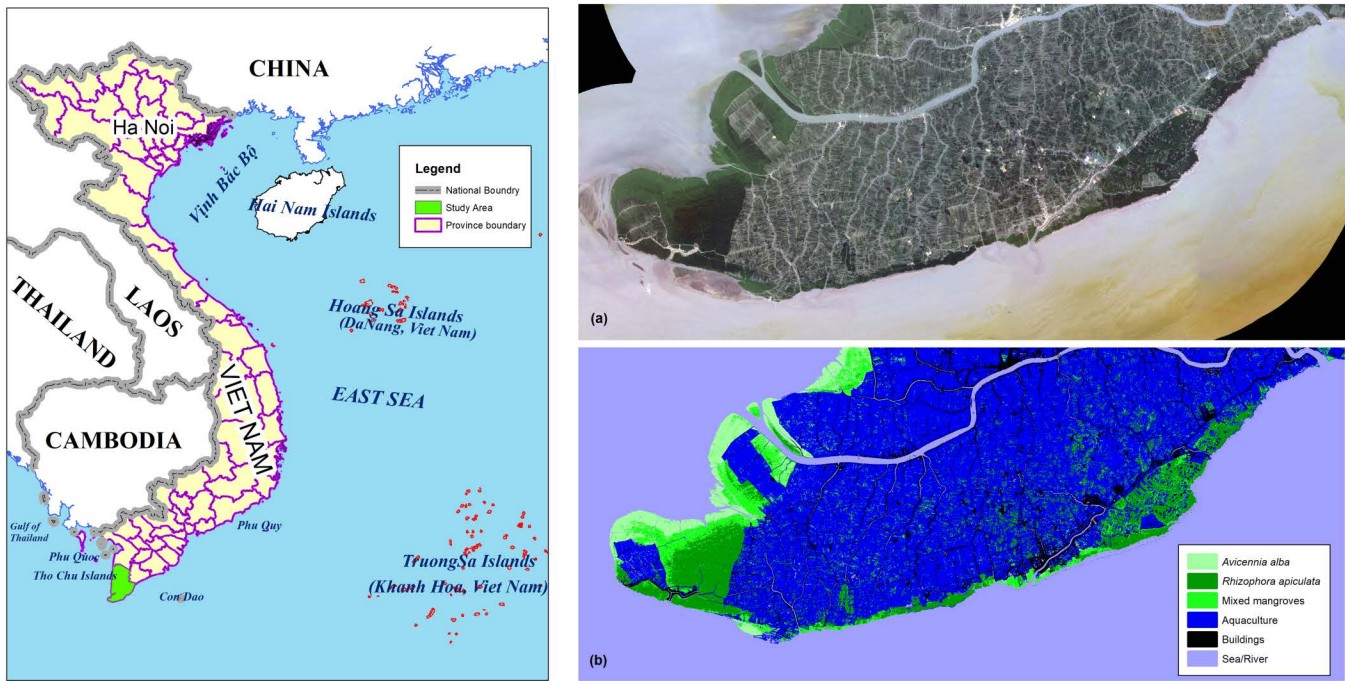

**Fig 2. Dataset: (a) Original image, (b) Labelled image.**

design with two repeated convolutions at each step, followed by a max-pooling operation. High-level features from the input image are extracted in this path. Afterwards, the expansive path boosts the spatial resolution of the image by up-sampling processes and concatenates it with relevant feature maps from the contracting path. Combining high-resolution and context-rich data enables the network to generate detailed segmentation maps. The final layer typically employs a 1x1 convolution to map each feature vector to the desired number of output classes. Moreover, different pre-trained models corresponding to different backbones (ResNet, VGG, or EfficientNet) provide a solid foundation for the U-Net. This improves its capability to handle complex images and makes it adaptable to different application domains, from biomedical imaging to satellite image analysis. The ability of a U-Net backbone model to extract intricate patterns and representations from the data is often increased by the number of backbone parameters. This results in higher accuracy if the model is appropriately trained and regularized [30]. Table 1 highlights the fundamental details of the backbones encountered in Google Colab, an outstanding environment for Python development and execution that gets rid of the requirement for local software installations, aside from importing necessary Python packages within the notebook environment.

In this study, to satisfy the objective of improving classification performance and prediction efficiency, in each backbone type, each backbone with the highest computational complexity (the maximum number of parameters, depth layers, and Giga Floating Point Operations per Second - GFLOPs) was selected to train. In addition, if two backbones of the same type have the same values for one of the criteria (number of parameters, depth, or GFLOPs) or have equivalent values for all criteria, both backbones were considered for the experiment. Twelve U-Net backbone models selected were Model 1 (M1): ResNet-152; Model 2 (M2): SEResNet-152; Model 3 (M3): SEResNeXt-101; Model 4 (M4): SENet-154; Model 5 (M5): ResNeXt101; Model 6 (M6): VGG19; Model 7 (M7): DenseNet201; Model 8 (M8): InceptionResNetV2; Model 9 (M9): Inception-v3; Model 10 (M10): MobileNet; Model 11 (M11): MobileNet-v2; Model 12 (M12): EfficientNet-B7.

**Table 1. Fundamental information about backbones.**

| No. | Types of backbone | Backbone | Year | Number of parameters | Depth (Layers) | GFLOPs (Input size: 224x224) |
|---|---|---|---|---|---|---|
| 1 | ResNet [51] | ResNet-18 | 2015 | 11.7 M | 18 | ~1.8 |
| 2 | | ResNet-34 | | 25.6 M | 34 | ~3.6 |
| 3 | | ResNet-50 | | 26 M | 50 | ~3.8 |
| 4 | | ResNet-101 | | 44.6 M | 101 | ~7.6 |
| 5 | | ResNet-152 | | 230 M | 152 | ~11.3 |
| 6 | SEResNet [42] | SEResNet18 | 2018 | 11.8 M | 18 | ~2.2 |
| 7 | | SEResNet34 | | 21.8 M | 34 | ~4.1 |
| 8 | | SEResNet50 | | 28.1 M | 50 | ~5.2 |
| 9 | | SEResNet101 | | 49.3 M | 101 | ~9.3 |
| 10 | | SEResNet152 | | 60 M | 152 | ~12.5 |
| 11 | SE-ResNeXt [42] | SE-ResNeXt50 | 2018 | 25 M | 50 | ~5.6 |
| 12 | | SE-ResNeXt101 | | 49 M | 101 | ~10.2 |
| 13 | SENet [42] | SENet154 | 2018 | 115.8 M | 154 | ~20.8 |
| 14 | ResNeXt [42] | ResNeXt 50 | 2017 | 25 M | 50 | ~4.6 |
| 15 | | ResNeXt101 | | 88 M | 101 | ~8.0 |
| 16 | VGG [52] | VGG-16 | 2014 | 138 M | 16 | ~15.5 |
| 17 | | VGG-19 | | 144 M | 19 | ~19.6 |
| 18 | DenseNet [53] | DenseNet-121 | 2016 | 8.1 M | 121 | ~2.9 |
| 19 | | DenseNet-169 | | 14.3 M | 169 | ~3.4 |
| 20 | | DenseNet-201 | | 20.2 M | 201 | ~4.4 |
| 21 | Inception | Inception-ResNet-V2 | 2015 | 21.8 M | 164 | ~13.2 |
| 22 | [54,55] | Inception-v3 | 2015 | 21.8 M | 48 | ~5.7 |
| 23 | MobileNet | MobileNet | 2017 | 4.2 M | 28 | ~0.6 |
| 24 | [44,56] | MobileNet-v2 | 2017 | 3.4 M | 53 | ~0.3 |
| 25 | EfficientNet [46] | EfficientNet B0 | 2020 | 5.3 M | 82 | ~0.39 |
| 26 | | EfficientNet B1 | | 7.8 M | 97 | ~0.7 |
| 27 | | EfficientNet B2 | | 9.2 M | 109 | ~1.0 |
| 28 | | EfficientNet B3 | | 12 M | 131 | ~1.8 |
| 29 | | EfficientNet B4 | | 19 M | 193 | ~4.2 |
| 30 | | EfficientNet B5 | | 30 M | 276 | ~9.9 |
| 31 | | EfficientNet B6 | | 43 M | 344 | ~19.0 |
| 32 | | EfficientNet B7 | | 66 M | 466 | ~37.0 |

## Image preprocessing and model setup

The experiment was carried out in Google Colab Pro using Python 3 and the Google Compute Engine backend with a GPU-T4, which has 40 GB of GPU RAM. The deep U-Net backbone models were trained to utilize the 8-bit original and labelled images. Three bands (R, G, and B) were comprised in the original image. The two images were divided into tiles of 256 × 256 pixels. Increasing the amount of training data enhances the resilience of network training and the quality of segmentation outcomes. The tiles were randomly divided into training samples of 75% and validation samples of 25% for model training. In the training process, a spectrum of colors ranging from black to white was illustrated in the tiles from the original image. The Adam optimization algorithm was adopted in the optimizer. The learning rate of the optimization algorithm directly affects the pace at which the network training process converges. The 12 backbones underwent pre-training on ImageNet, configuring the activation function as softmax. One hundred epochs with eight batches per epoch were

**Table 2. Experimental results under three loss employment cases: (1) Focal loss, (2) Dice loss, and (3) the combination of Focal loss and Dice loss.**

| Classes | Metrics (%) | Models M3 | | |
|---|---|---|---|---|
| | | Focal loss | Dice loss | Focal loss and Dice loss |
| All classes | IoU | 75.59 | 74.54 | 76.63 |
| | $F_1$-score | 85.26 | 84.74 | 85.91 |
| | OA | 92.21 | 90.66 | 93.17 |

implemented. The value of the epoch was used to ensure the precision and convergence of the loss. It would ascertain the model's performance and impact the duration of network training.

Loss function, the combination of Focal loss and Dice loss, was adopted to evaluate the training performance after comparing experimental results under three loss employment cases: Focal loss (2), Dice loss (5), and the combination of Focal loss and Dice loss (1). The combination of Focal loss and Dice loss marked an outstanding identification for all classes, which gained the highest quantitative assessment metrics (Table 2). Focal loss is a modified version of the conventional cross-entropy loss that explicitly tackles the problem of class imbalance, where the number of positive samples (objects of interest) is much lower than the number of negative samples (background) [31]. In other words, poor performance results from the model's tendency to ignore the positive samples and concentrate only on the negative ones. This problem is solved by the Focal loss, which up-weights the complicated positive samples and down-weights the simple negative samples. Besides, the similarity between the predicted segmentation mask and the mask from ground truth data is assessed using Dice loss, also known as the Dice similarity coefficient and written using the definition of precision (3) and recall (4) [32]. It is the most widely used segmentation evaluation metric and directly optimizes.

$$Total\ loss = Focal\ loss + Dice\ loss \tag{1}$$

$$FocalLoss\ (gt,\ pr) = -gt * \alpha * (1 - pr)^{\gamma} * \log{(pr)} \tag{2}$$

$$Precision = \frac{TruePositive}{TruePositive + FalsePositive} \tag{3}$$

$$Recall = \frac{TruePositive}{TruePositive + FalseNegative} \tag{4}$$

$$DiceLoss\ (Precision,\ Recall) = 1 - (1 + \beta^2\ )\frac{Precision * Recall}{\beta^2 * Precision * Recall} \tag{5}$$

Where $gt$ is ground truth label (1 for positive class, 0 for negative class), $pr$ is predicted probability for the positive class, $\alpha$ value is balancing factor for handling class imbalance and taken as 0.25, $\gamma$ value is focusing parameter to reduce the loss for easy examples and taken as 2, and $\beta$ value is a weight factor controlling the trade-off between precision and recall and taken as 1 to have the maximum accuracy rate of the U-Net model.

## Accuracy assessment

In this study, the primary metrics used to assess the performance of trained U-Net backbone models for land cover or mangrove species classification were overall accuracy (OA) (6), $F_1$-score (7), and intersection over union (IoU) (8). However, the dataset used was imbalanced, where six classes have different numbers of pixels (samples). The IoU and $F_1$-score were more informative and gained more attention because they focused on the overlap between prediction and

growth truth and ensured that the model detected minority classes effectively. The OA was only used to support quick model comparisons alongside metrics of IoU and $F_1$-score. In particular:

OA was determined by summing the percentages of pixels that were accurately identified by the model in comparison to the reference labelled image for all categories. The accuracy rate quantifies the number of accurate pixel predictions [13].

$$OA = \frac{True\ Possitive + True\ Negative}{True\ positive + True\ Negative + False\ Positive + False\ Negative} \quad (6)$$

$F_1$-score was a metric that integrates precision (the ability of the model to correctly identify positive samples) and recall (the ability of the model to identify all positive examples in the dataset) to produce a single value that quantifies the overall performance of a classification model. A higher $F_1$-score demonstrates that the model achieves a better balance between precision and recall, while a low $F_1$-score implies that the model may excel in either precision or recall but not both simultaneously [31]. It was particularly advantageous in situations as our experiment where classes are unevenly distributed.

$$F_1 - score = 2\ \times\ \frac{Precision\ \times\ Recall}{Precision + Recall} \quad (7)$$

IoU, typically referred to as Jaccard Index, is a commonly used performance assessment statistic in semantic segmentation or object identification [31,33]. The IoU metric was computed by dividing the intersection of the predicted image and reference image by combining the predicted image and reference image. A high IoU value indicates that the predicted image closely aligns with the reference image. In contrast, a low IoU value suggests a significant deviation between the predicted and the reference images [31]. The IoU value was computed individually for each class in this study. By taking the average of these values, the average IoU value was obtained for all model classes.

$$IoU = \frac{Target\ \cap\ Prediction}{Target\ \cup\ Prediction} \quad (8)$$

## Proposed ensemble U-Net model

An ensemble Model (EM) is used to merge predictions derived from several fundamental models to mitigate excessive variability and bias [22]. In particular, a model may exhibit high performance in some classes while demonstrating low performance in others. In ensemble learning, the combination of several models allows for the accurate classification of characteristics that one model inadequately learned by using the patterns acquired from other models. To verify the increase in efficiency of mangrove species classification or land-cover classification, the study proposed testing with the integration of three single U-Net backbone models having the best evaluated OA indices. Various techniques exist for constructing an ensemble model, and the weighted averaging ensemble approach was used in this work. A weighted ensemble is an advancement of a model-averaging ensemble, where the model's performance determines the weight of each member's contribution to the final prediction. The high-performing model has more weight than the low-performing model [22,34]. The final equation is given in (9).

$$P(t) = \sum_{i=1}^{N} w_i p_i(t) \quad (9)$$

Where N is the total number of models, $p_i$ is the probability for the model i, and $w_i$ is the weight of each model.

## Research results

### Single model evaluation and ensemble model selection

Twelve single U-Net backbone models were trained on Google Colab with the same input dataset and set-up parameters. It can be seen in Fig 3, training and validation accuracy generally rose as the number of epochs grew, whereas loss decreased. For most of the trained models, there was a quick improvement in accuracy and a drop in loss in the first stages, followed by a moderate and stable progression after around 15 epochs. Besides, the number of parameters, depth, and GFLOPs demonstrated in Table 1 affected the computation complexity of any single U-Net backbone model. The higher the values of these criteria were, the more the computation complexity and time training (~20 minutes to ~19 hours) increased.

Our study aimed at not only classifying mangrove species but also ensuring that the remaining classes were well classified, with high accuracy. Thereby, for the next step of developing an ensemble U-Net model, the single U-Net backbone model results were analyzed based on the assessment results of all classes. Regarding the quantitative evaluation of the trained models, it can be witnessed in Fig 4 that the values of OA, IoU, and $F_1$-score experienced a similar trend for all single models. In other words, when comparing the evaluation results of two single models, all three quantitative assessment metrics were simultaneously greater than the ones of the remaining model, or vice versa. On the validation set, when the models achieved their training performance, all U-net backbone models witnessed good OA accuracies (>50%) [33]. Almost all models had OA, IoU, and $F_1$-score in the range of around 86–93%, 68–77%, and 80–87%. Out of twelve models, there were three models with outstanding results of OA: M3 (93.17%), M10 (93.31%), and M12 (92.23%). On the training set, the accuracy rates of IoU for the three best models were 76.63%, 76.72%, and 75.91%, respectively, while $F_1$-score were 85.91%, 86.66%, and 85.59%, respectively. When the models were fully trained, the M10 exhibited the highest classification accuracy with all three metrics. Besides, the M7 (67.9%) and M11 (66.4%) models experienced lower OA values than the others. As a result, three U-Net backbone models (M3 - SEResNeXt-101, M10 - MobileNet, M12 - MobileNet) were chosen for our proposed ensemble model (EM) to improve the performance results of land cover comprising mangrove species classification.

### Quantitative evaluation of ensemble model

The accuracy rate measure was prone to bias regarding differentiating classes, mainly when the background dominated the majority of the image. Hence, the average IoU statistic was more significant when assessing the effectiveness of semantic segmentation. The best performance of the ensemble decision was obtained based on maximum IoU (80.08%) with the coefficients of 0.2xM3 + 0.0xM10 + 0.1xM12. In other words, the proposed ensemble model was a combination of M3 and M12 with a ratio of 2:1. The detailed evaluation of the six classes of the four models (M3, M10, M12, and EM) using quantitative metrics (OA, IoU, and $F_1$-score) was pointed out in Table 3.

As can be seen, the proposed EM model outperformed the typical single-component U-Net backbone models (M3, M10, and M12). The rates of IoU, $F_1$-score, and OA were 80.08%, 95.82%, and 95.9%, respectively. Regarding the quantitative evaluation for each class, the two classes (aquaculture and sea/river) of the EM exhibited more outstanding IoU and $F_1$-score compared to the single-component models. The values of IoU and $F_1$-score of these two classes were higher than 96%, in which the sea/river class had an IoU of 98.49% and a $F_1$-score of 99.24%. Besides, the assessment values of the remaining four classes, including three species classes, demonstrated better and more consistent accuracy. The IoU and $F_1$-score of *Avicennia alba*, *Rhizophora apiculate*, and mixed mangroves fluctuated in a range of 64%−80% for IoU values and 78–89% for $F_1$-score values, which were of good accuracy.

### Visual classification evaluation of ensemble model

In terms of the visual prediction, it is apparent in Fig 5 that a better comprehension of the EM model's classification performance was provided. Dark blue, navy blue, light blue, green, orange and brown were respectively represented colors

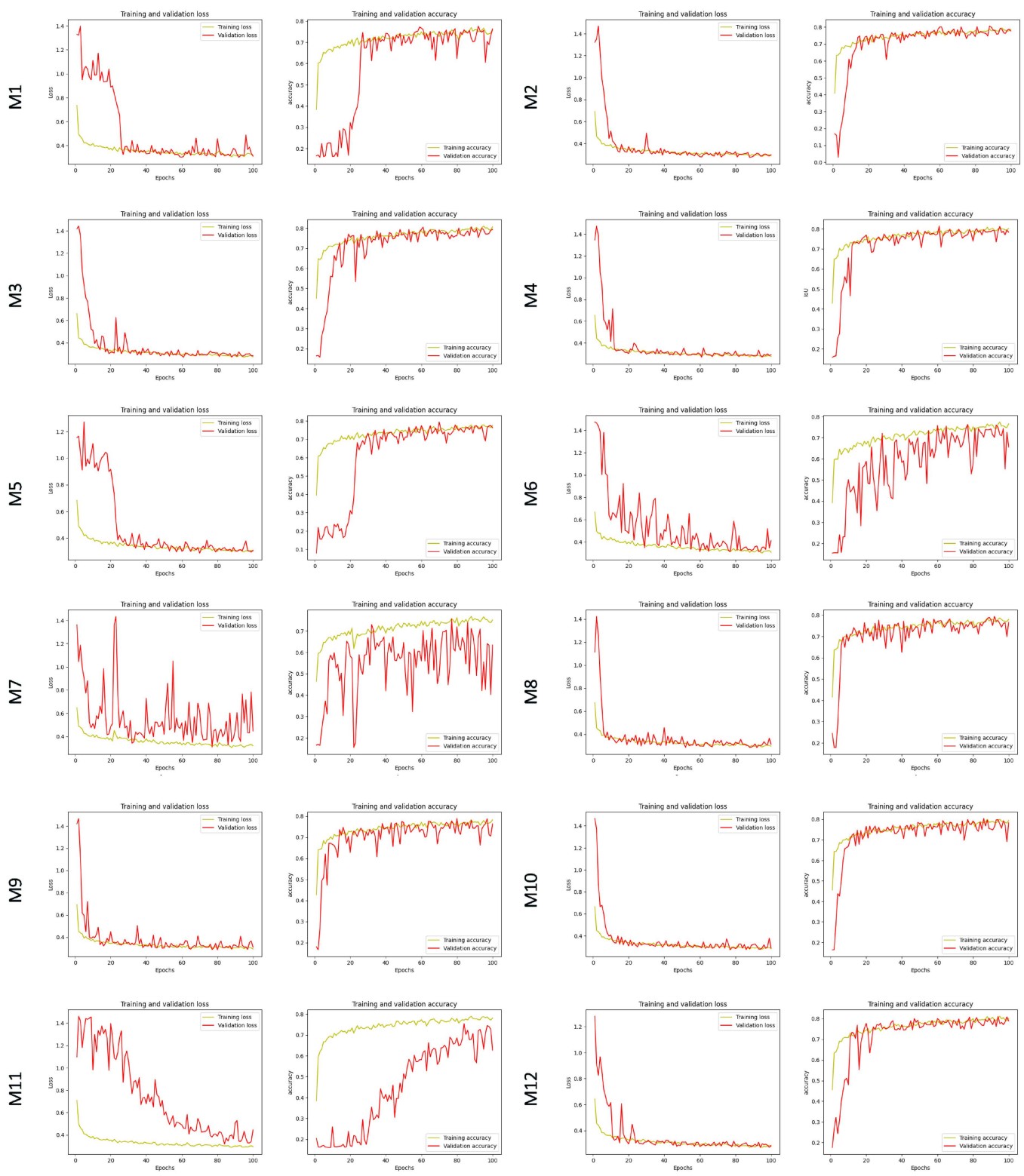

**Fig 3. The training processes of deep learning U-Net backbone models.**

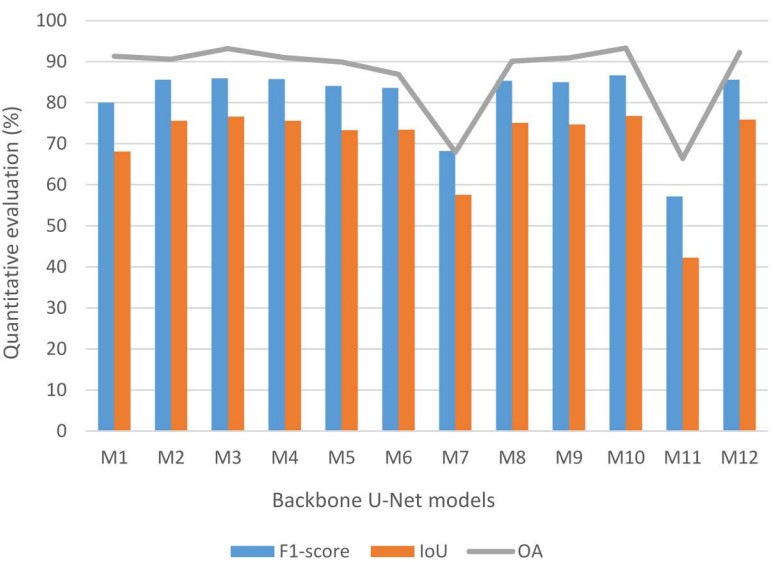

**Fig 4. Quantitative evaluation of U-Net backbone models.**

**Table 3. Sample classification results with different deep-learning U-Net backbone models.**

| Classes | Metrics (%) | Models | | | | | | | | | | | | |
|---|---|---|---|---|---|---|---|---|---|---|---|---|---|---|
| | | M1 | M2 | M3 | M4 | M5 | M6 | M7 | M8 | M9 | M10 | M11 | M12 | EM |
| *Avicennia alba* | IoU | 46.96 | 89.70 | 85.90 | 66.28 | 66.97 | 67.78 | 92.31 | 80.36 | 64.16 | 76.86 | 9.70 | 55.79 | 76.01 |
| | $F_1$-score | 63.91 | 94.57 | 92.41 | 79.72 | 80.22 | 80.80 | 96.00 | 89.11 | 78.17 | 86.91 | 17.69 | 71.62 | 86.37 |
| *Rhizophora apiculata* | IoU | 70.82 | 63.68 | 70.55 | 78.91 | 73.38 | 90.76 | 80.34 | 57.57 | 77.93 | 68.07 | 37.81 | 88.47 | 79.78 |
| | $F_1$-score | 82.92 | 77.81 | 82.73 | 88.21 | 84.64 | 95.15 | 89.10 | 73.07 | 87.60 | 81.00 | 54.88 | 93.88 | 88.75 |
| Mixed mangroves | IoU | 56.47 | 69.34 | 51.63 | 62.37 | 59.41 | 44.84 | 68.87 | 63.54 | 60.48 | 77.06 | 52.40 | 62.30 | 63.53 |
| | $F_1$-score | 72.18 | 81.89 | 68.10 | 76.82 | 74.53 | 61.92 | 81.56 | 77.70 | 75.37 | 87.04 | 68.77 | 76.77 | 77.70 |
| Aquaculture | IoU | 91.61 | 90.80 | 91.91 | 88.74 | 88.06 | 85.41 | 66.69 | 88.24 | 88.92 | 88.33 | 60.14 | 91.06 | 92.59 |
| | $F_1$-score | 95.62 | 95.17 | 95.78 | 94.03 | 93.65 | 92.13 | 80.02 | 93.75 | 94.14 | 93.80 | 75.11 | 95.32 | 96.15 |
| Buildings | IoU | 58.68 | 63.77 | 66.69 | 68.53 | 63.57 | 65.42 | 19.11 | 77.52 | 67.68 | 69.86 | 38.42 | 70.63 | 70.09 |
| | $F_1$-score | 73.96 | 77.88 | 80.02 | 81.32 | 77.73 | 79.09 | 32.09 | 87.33 | 80.72 | 82.25 | 55.51 | 82.79 | 82.41 |
| Sea/River | IoU | 84.10 | 76.19 | 93.07 | 88.90 | 88.27 | 86.23 | 18.10 | 83.42 | 88.88 | 80.14 | 55.09 | 87.23 | 98.49 |
| | $F_1$-score | 91.36 | 86.48 | 96.41 | 94.12 | 93.77 | 92.60 | 30.65 | 90.96 | 94.11 | 88.97 | 71.04 | 93.17 | 99.24 |
| All classes | IoU | 68.11 | 75.58 | 76.63 | 75.62 | 73.28 | 73.41 | 57.57 | 75.11 | 74.68 | 76.72 | 42.26 | 75.91 | 80.08 |
| | $F_1$-score | 79.99 | 85.63 | 85.91 | 85.71 | 84.09 | 83.61 | 68.24 | 85.32 | 85.02 | 86.66 | 57.17 | 85.59 | 95.82 |
| | OA | 91.27 | 90.59 | 93.17 | 90.95 | 89.92 | 86.91 | 67.90 | 90.11 | 90.88 | 93.31 | 66.40 | 92.23 | 95.90 |

for six classes (*Avicennia alba*, *Rhizophora apiculata*, mixed mangroves, aquaculture, buildings, and sea/river). The deep learning EM was proficient in correctly classifying remote sensing images concerning the overall visual impact. Predicted images from the EM closely matched the labels from the ground reality. For the prediction of images 2 (2d) and 3 (3d), the classes River, *Rhizophora apiculata*, mixed mangroves, and buildings had better predictions than testing labels. Nevertheless, variations arise in the specific predictions generated by the model. The prediction results from images 1 and 4 show that the road class (or we grouped it as the buildings class) was predicted as aquaculture. This was also considered an error from the classified land-cover image (or labelled image) we classified from the input.

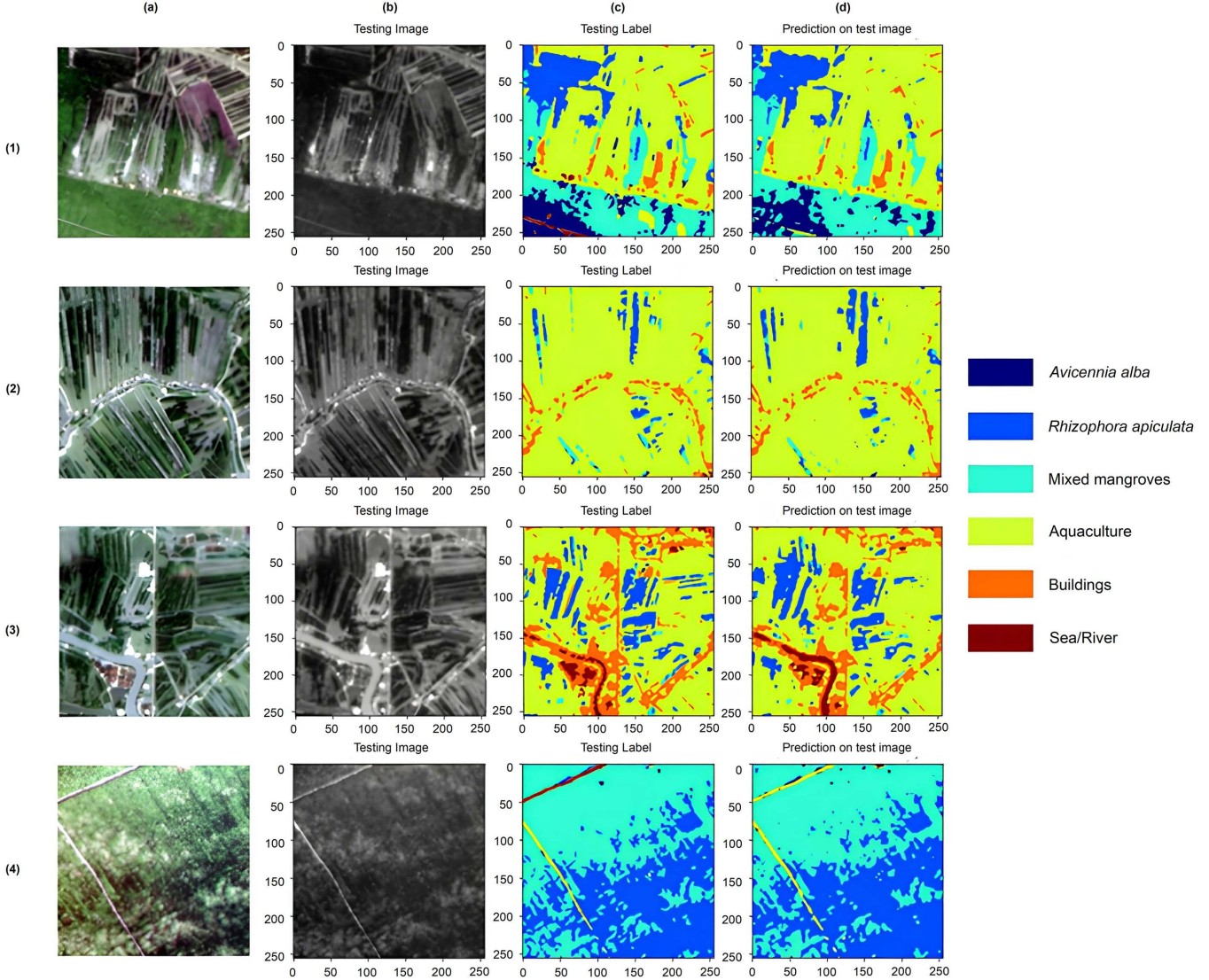

**Fig 5. Classification diagram of ensemble model: (a) Patchified original image, (b) Patchified trained image/testing image, (c) Patchified labelled image/testing label, and (d) Prediction based on testing image.**

## Discussion

### Single U-Net backbone model vs conventional classification method for mangrove classification

In this experiment, twelve single U-Net backbone models were successfully trained. The findings indicated that the U-Net backbone models exhibited superior performance compared to traditional classification methods and conventional machine learning approaches for the purpose of land-cover classification, explicitly considering mangrove areas. The conventional approaches were restricted due to various constraints (the difficulty in sampling design and collection for ground data acquisition over a large scale, the need for users' experiences and expertise in the classification procedures, problems related to spectral aspects at a given resolution, or challenging to automate), thereby gaining not outstanding classification results and costly and time-consuming drawbacks. For instance, the typical research conducted by Lu et al.

(2004) [35] indicated that minimum-distance classifier (MDC) led to notable differences in the variance of the classes and misclassification. Additionally, the classification abilities of extraction and classification of homogeneous objects (ECHO) and a decision-tree classifier based on linear spectral mixture analysis (DTC-LSMA) were found to be most effective for mature forests rather than newly plants. Machine learning methods were applied because of the independence of data distribution assumptions, making them more precise and effective than traditional classification methods in high-dimensional and complicated data spaces. However, numerous machine learning methods, such as support vector machines, provided complexity owing to the extensive range of parameters that needed adjustment and were challenging to automate [36–39].

On the other hand, the U-Net deep learning model has numerous merits in overcoming the disadvantages of the traditional approaches. Without requiring manual feature extraction or domain-specific expertise, the U-Net model enables the capture of intricate spatial patterns. Hence, U-Net is very beneficial for large-scale, dynamic land-cover mapping because it provides a solid and adaptable environmental monitoring and management solution. According to Liu et al. (2020) [29], changes in land cover over time were monitored and categorized effectively by the U-Net model since it could incorporate multi-temporal satellite images, an advantage over traditional algorithms that may struggle with handling temporal data. Additionally, the U-Net model, with its advanced deep learning architecture, has the ability to learn from satellite images directly. In addition to spectra, the color and structure of objects in satellite images were also attractive. This was proven to be extremely helpful in classifying mangrove species in our study, where the classification accuracy of species was almost more significant than 60%. Furthermore, the U-Net model's ability to accurately segment was enhanced by using pre-trained encoders, hence decreasing the need for extensive labelled datasets. The inclusion of residual units and rich skip connections in the network could simplify the training of deep networks while boosting the transmission of information. This enables the creation of networks with fewer parameters and improved performance, as shown by Zhang et al. (2018) [40]. These remarks elucidated the exceptional classification outcomes achieved by the majority of the U-Net backbone models, with an overall accuracy (OA) ranging from over 86% to over 93%. In another study conducted in 2024, Hao et al. [13] investigated the growing significance of high-resolution imagery obtained from unmanned aerial vehicles (UAVs) using deep learning models, including FCN-8s, SegNet, U-net, and Swin-UNet, in land-use mapping. They found that U-Net attained an overall accuracy of 91.90%. Conversely, the results of Ma et al. (2025) [41] demonstrated that traditional and machine learning classifiers obtained only moderate accuracy (approximately 60–69%) when applied individually to high-quality composite Landsat imagery despite leveraging robust classifiers such as support vector machine (SVM), random forest (RF), or gradient tree boost (GTB). In Hai et al.'s research (2022) [3], the accuracy (OA) achieved for categorizing mangrove forests in Mui Ca Mau, Vietnam, using the random forest (RF) technique, was only 80%. Our findings were approximately 10% greater than this finding. These demonstrations emphasize the demand for more advanced U-Net models in high-complexity, class-imbalanced applications, including mangroves.

### Proposed ensemble model for mangrove species classification

The combination of three individual U-Net backbone models to propose an ensemble U-Net model resulted in a slight improvement in the recognition of mangrove species, and a significant increase in the overall accuracy of land-cover classification for all classes including mangroves. The three most optimal single models for proposing our EM were well-recognized and proven in various domains for image classification. First, SEResNeXt-101 combines the strengths of ResNeXt and 'Squeeze-and-Excitation' (SE) blocks, leading to enhance the network's sensitivity to relevant features by dynamically recalibrating channel-wise feature responses [42]. Although more parameters and expensive computation were experienced in SeResNeXt-101, good results were demonstrated on the ImageNet classification tasks [43]. This would result in a model that was particularly adept at distinguishing subtle differences in land-cover types, making it invaluable for complex remote sensing tasks like forest cover classification, urban area detection, and agricultural monitoring. In a study on recognizing rice diseases by CNN-based deep learning architectures, Ahad et al. (2023) [43] found

that SEResNeXt-101 helped improve by 17% accuracy after a transfer learning process. Second, in terms of MobileNet, its lightweight architecture, based on depth-wise separable convolutions, significantly reduces the number of parameters and computational cost, enabling real-time classification on edge devices or in field conditions [44]. This makes MobileNet a preferred choice for real-time land-cover monitoring and disaster response tasks. Gyasi and Swarnalatha (2017) [45] stated that the Cloud-MobiNet model, built by deploying MobileNet as a basis, achieved an accuracy success rate of about 98% in classifying ground-based clouds. Given that automated ground-based cloud categorization was expected to be the preferred approach of cloud observation in meteorological research and prediction, as well as in the aviation and aeronautical sectors, Cloud-MobiNet could become an essential model in the near future. Lastly, a seminal paper, "EfficientNet: Rethinking Model Scaling for Convolutional Neural Networks" by Tan and Le (2019) [46], described how EfficientNets exceeded previous convolutional neural networks such as ResNet or Inception in terms of accuracy and efficiency. Third, using a compound scaling technique, EfficientNet-B7 achieved state-of-the-art accuracy while maintaining efficiency by optimally balancing network depth, width, and resolution. Its ability to scale effectively ensures that it can be adapted for various levels of image complexity, providing robust performance across different remote sensing scenarios.

As a result, when our experiment suggested the EM combining the optimum characteristics of the single-component models, the classification accuracy rose by 3%, and the consistency of the classes' accuracy improved. Our result was consistent with that of Sevi and Aydin (2023) [33] in detecting railway lines by applying U-Net segmentation performance using an ensemble model. Besides, regarding performance interpretation, the poorer classification accuracy of mangrove species classes compared to the other land-cover classes (aquaculture, buildings, and sea/rivers) might be attributed to the input dataset used in training. In the input dataset, the accuracy of recognizing mangrove species in the labelled image was not exceedingly good. Furthermore, the mangrove forest area only accounted for around a quarter of the entire area. Plus, when evaluating the performance of a classification model, it is essential to consider multiple metrics to gain a comprehensive understanding [47–49]. High OA might suggest that the model was performing well. Besides, the low IoU or $F_1$-score indicated an underlying issue with class imbalance. Specifically, the model could be proficient at identifying the majority class, which occurred more frequently in the dataset but struggled to classify the minority class, which was less represented correctly. In scenarios with imbalanced data, the OA could be misleadingly high because the model predominantly predicted the majority class, inflating the overall accuracy while neglecting the minority class. Low IoU or $F_1$-score highlighted this deficiency, revealing that the model failed to capture the nuances of less frequent classes, leading to poor performance in identifying these classes despite a high OA. This underscored the importance of using complementary metrics like IoU and $F_1$-score to accurately assess a model's ability to classify all classes effectively, not just the most common ones.

## Limitations and future extensions

Besides the remarkable results achieved, our research still has limitations that need to be further studied and overcome in the future. First of all, Planet imagery, despite its high spatial resolution (5 m), has more constricted spectral bands than Radar, LiDAR or UAV data, which may limit the improvement of the model's capacity to discriminate mangrove species with similar spectral signatures. This demonstrated that the quantitative assessment metrics of the EM did not increase over 80% of IoU and 90% of $F_1$-score. Secondly, our input dataset depended mainly on a large satellite image of a specific area (Ngoc Hien, Ca Mau, Vietnam), so the number of samples of specific mangrove species was underrepresented in the training dataset. This led to class imbalance issues, and the influence of the model's generalizability, particularly for less dominant mangrove species. Thirdly, while the ensemble U-Net approach boosted model accuracy, it also increased computational demands, which made it challenging to implement for resource-constrained applications or real-time monitoring.

Considering the aforementioned constraints, the following are some potential avenues for future study to tackle the issues. By offering richer spectral and structural information, combining high-resolution imagery (Planet or World-View) with other remote sensing data (such as Sentinel, SAR, LiDAR, or UAV-based hyperspectral photography) may improve classification accuracy. The study conducted by Irfan et al. (2025) [50] elucidated the synergistic possibilities of

SAR-optical data fusion for specific land type categorization, utilizing the SEN12MS benchmark dataset to provide significant texture and structural details. Moreover, adding more samples of mangrove species to the input data is also an effective solution to increase training efficiency. Also, experiments combining other single U-Net backbone models that are lightweight and efficient might be conducted to reduce computational costs and maintain high accuracy. Last but not least, the transfer deep learning techniques (the M10 model or our proposed ensemble U-Net model) could be employed to adapt to different geographic regions and improve its generalization to other mangrove areas. Expanding the study to incorporate multi-temporal data is promised to enable tracking of mangrove growth, degradation, and species succession, strengthening conservation and management strategies.

## Conclusion

The research experiment effectively trained individual U-Net backbone models and introduced an integrated model that achieves more than 95% accuracy for land-cover classification, including mangrove species. A small dataset and medium-quality satellite imagery could also obtain high classification accuracy. The suggested ensemble model could establish a land-cover map that includes mangrove species. Meanwhile, the MobileNet model enables the development of a distribution map specifically for mangrove species. Although certain constraints remain in the study, regarding the input data or single models chosen for the ensemble one, this also offers up novel fields of study for the authors to explore deeper. In the future, single U-Net backbone or ensemble models are expected to replace conventional classification models and adapt for different geographic regions in monitoring and managing mangrove ecosystems.

## Supporting information

**S1 File. Code for the proposed ensemble model.**
(IPYNB)

**S2 File. Code for the single U-Net model (Model 12).**
(IPYNB)

**S3 File. Code for the single U-Net model (Model 11).**
(IPYNB)

**S4 File. Code for the single U-Net model (Model 10).**
(IPYNB)

**S5 File. Code for the single U-Net model (Model 9).**
(IPYNB)

**S6 File. Code for the single U-Net model (Model 8).**
(IPYNB)

**S7 File. Code for the single U-Net model (Model 7).**
(IPYNB)

**S8 File. Code for the single U-Net model (Model 6).**
(IPYNB)

**S9 File. Code for the single U-Net model (Model 5).**
(IPYNB)

**S10 File. Code for the single U-Net model (Model 4).**
(IPYNB)

**S11 File. Code for the single U-Net model (Model 3).**
(IPYNB)

**S12 File. Code for the single U-Net model (Model 2).**
(IPYNB)

**S13 File. Code for the single U-Net model (Model 1).**
(IPYNB)

**S14 File. Code for testing losses.**
(IPYNB)

**S15 File. Input dataset.**
(ZIP)

## Acknowledgments

We thank the Vietnam Ministry of Agriculture and Environment (project TNMT.ĐL.2023.04) for supporting our study. In addition, we would like to express our gratitude to the reviewers and academic editor for their constructive comments that significantly enhanced this manuscript.

## Author contributions

**Conceptualization:** Tran Dang Hung, Minh Hai Pham, Tran Hong Hanh.

**Data curation:** Tran Dang Hung, Minh Hai Pham, Bui Thanh Huyen, Tran Hong Hanh.

**Formal analysis:** Tran Dang Hung, Bui Thanh Huyen.

**Investigation:** Tran Dang Hung, Pham Hong Tinh.

**Methodology:** Tran Dang Hung, Minh Hai Pham, Bui Thanh Huyen.

**Project administration:** Tran Dang Hung.

**Supervision:** Minh Hai Pham, Tran Hong Hanh.

**Validation:** Tran Dang Hung, Minh Hai Pham, Bui Thanh Huyen, Pham Hong Tinh, Nguyen Thanh Bang, Tran Thanh Tung.

**Visualization:** Tran Dang Hung, Bui Thanh Huyen.

**Writing – original draft:** Tran Dang Hung, Minh Hai Pham, Bui Thanh Huyen, Pham Hong Tinh, Nguyen Thanh Bang.

**Writing – review & editing:** Tran Dang Hung, Minh Hai Pham, Bui Thanh Huyen.

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
