## [Decision Letter · Decision Letter 0]

PONE-D-24-60427Mangrove species classification using a proposed ensemble U-Net model and Planet satellite imagery: A case study in Ngoc Hien district, Ca Mau province, VietnamPLOS ONE

Dear Dr. Pham,

Thank you for submitting your manuscript to PLOS ONE. After careful consideration, we feel that it has merit but does not fully meet PLOS ONE’s publication criteria as it currently stands. Therefore, we invite you to submit a revised version of the manuscript that addresses the points raised during the review process.

Please revise the manuscript provided at the link above in accordance with the reviewers' comments and upload the revised version. Kindly take note of the following checklist:

Ensure that all references cited are directly relevant to the manuscript's content.Clearly highlight any revisions made to the manuscript so that editors and reviewers can easily identify the changes.Submit a cover letter addressing the reviewers' comments, detailing your responses point by point, and explaining the revisions made to the manuscript.If the reviewers suggested additional references, critically assess their relevance and include them only if they enhance the quality of your manuscript. If you consider the suggested references unnecessary, you may omit them.If there are comments in the review reports that you found impossible to address, provide a detailed explanation in your response.

We look forward to receiving your revised manuscript.

Kind regards,

Rodrigo Nogueira Vasconcelos, Ph.D.

Academic Editor

PLOS ONE

2. We note that your Data Availability Statement is currently as follows: [All relevant data are within the manuscript and its Supporting Information files.] Please confirm at this time whether or not your submission contains all raw data required to replicate the results of your study. Authors must share the “minimal data set” for their submission. PLOS defines the minimal data set to consist of the data required to replicate all study findings reported in the article, as well as related metadata and methods (https://journals.plos.org/plosone/s/data-availability#loc-minimal-data-set-definition).

3. Please ensure that you refer to Figure 2 in your text as, if accepted, production will need this reference to link the reader to the figure.

Additional Editor Comments:

Dear Authors,

Your manuscript has been reviewed by experts in the field and has received a "Major Review" classification. We kindly request that you carefully evaluate the reviewers' comments and implement the necessary revisions to address their feedback.

Reviewers' comments:

Reviewer's Responses to Questions

**Comments to the Author**

1. Is the manuscript technically sound, and do the data support the conclusions?

Reviewer #1: Yes

Reviewer #2: Partly

2. Has the statistical analysis been performed appropriately and rigorously? 

Reviewer #1: I Don't Know

Reviewer #2: N/A

3. Have the authors made all data underlying the findings in their manuscript fully available?

Reviewer #1: Yes

Reviewer #2: No

4. Is the manuscript presented in an intelligible fashion and written in standard English?

Reviewer #1: Yes

Reviewer #2: Yes

5. Review Comments to the Author

Reviewer #1: The manuscript titled “Mangrove species classification using a proposed ensemble U-Net model and Planet satellite imagery: A case study in Ngoc Hien district, Ca Mau province, Vietnam” presents an interesting work on Mangrove species classification. However, the manuscript can be further improved in respect of the following points:

1. In line 28 on page 2, it should be “Intersection over Union”.

2. The literature survey can be improved focusing particularly on the work carried out for Mangrove species classification.

3. There is no mention about how the ground truth data was obtained.

4. Experiments can be carried out to prove the point of using a combination of Focal loss and Dice loss by assessing the impact of employing only the Focal loss or the Dice loss on the classification results. These results can be compared to those achieved using the combination of Focal loss and Dice loss.

5. What about the computational complexity of the models employed?

6. The quality of Figures 3, 4, and 5 needs to be improved.

7. In Table 2, the results for only four models, i. e., M3, M10, M12, and EM were shown, what about the other models?

Reviewer #2: Plant species identification using satellite images has been a widely addressed area of research. The novelty of this research is addressing a specific species, Mangrove, as a case study in Vietnam.

The following improvements are suggested.

1. It would be better to highlight the main research questions in the introduction section, and describe in the Discussion section, how you fulfilled those in the study, by referring to the applied methods and the obtained results.

2. Include a new section 2 for related studies. it would be better to consider ML based land use identification using satellite images published in recent years with the latest technologies. Discuss the techniques used in the related studies on land cover identification such as https://doi.org/10.1145/3056662.3056681 and https://doi.org/10.1002/9781119682042.ch1 together with the limitations in the existing studies and justify the proposed method.

You can consider other cropland identification work such as https://doi.org/10.1016/B978-0-323-85214-2.00009-4 and https://doi.org/10.1016/B978-0-323-90550-3.00008-4

3. Clearly describe the novel scientific contribution. There are many U-net types available in the literature. Justify the selection of your approach with ensemble U-Net.

4. Your scientific contribution is sufficient. However, you need to justify the contributions.

5. Most of the figures are not clear. You have to refine the visualization aspects of your study.

6. Clearly state the used datasets and the number of images for different labels, before augmentation and the data fed to the training process. – That is, the considering the two labels buildings/ background, the number of images in each class for training, testing and validation sets. Discuss data balancing/ imbalancing.

7. Have you check the generalizability of the proposed model, for different datasets.

8. Consider the results graphs/ figures. Instead of stating as Fig. x shows…it would be better to explain the observations that can be seen in the graph, and discuss the conclusion that can be made from those graph observations. This will help readers to identify the importance of the findings.

9. In the discussion section, justify the achievements of the said contributions or research questions mentioned in the introduction, referring to the followed methodology and the obtained results.

10. Compare the results of this study with latest related work.

11. In the discussion, state the research limitations and the future possible extensions.

12. Discuss the possible practical applications, of this model.

6. PLOS authors have the option to publish the peer review history of their article (what does this mean? ). If published, this will include your full peer review and any attached files.

**Do you want your identity to be public for this peer review?** For information about this choice, including consent withdrawal, please see our Privacy Policy .

Reviewer #1: **Yes: ** Preetpal Kaur Buttar

Reviewer #2: No

---

## [Author Response · Author response to Decision Letter 1]

4 Mar 2025

Dear Reviewers,

Thank you very much for your valuable comments. We have carefully checked the manuscript and revised as your comments. We have uploaded a new copy of better quality Figures as requested. We hope that our manuscript is now suitable for further processing.

---

## [Decision Letter · Decision Letter 1]

PONE-D-24-60427R1Mangrove species classification using a proposed ensemble U-Net model and Planet satellite imagery: A case study in Ngoc Hien district, Ca Mau province, VietnamPLOS ONE

Dear Dr. Minh Hai Pham,

Thank you for submitting your manuscript to PLOS ONE. After careful consideration, we feel that it has merit but does not fully meet PLOS ONE’s publication criteria as it currently stands. Therefore, we invite you to submit a revised version of the manuscript that addresses the points raised during the review process.

Based on the reviewers' evaluations, minor improvements are needed to enhance the manuscript. One reviewer also indicated that further improvement is still required. Therefore, the decision is for a minor revision.

We look forward to receiving your revised manuscript.

Kind regards,

Rodrigo Nogueira Vasconcelos, Ph.D.

Academic Editor

PLOS ONE

Journal Requirements:

Reviewers' comments:

Reviewer's Responses to Questions

**Comments to the Author**

1. If the authors have adequately addressed your comments raised in a previous round of review and you feel that this manuscript is now acceptable for publication, you may indicate that here to bypass the “Comments to the Author” section, enter your conflict of interest statement in the “Confidential to Editor” section, and submit your "Accept" recommendation.

Reviewer #2: All comments have been addressed

Reviewer #3: All comments have been addressed

2. Is the manuscript technically sound, and do the data support the conclusions?

Reviewer #2: Yes

Reviewer #3: Yes

3. Has the statistical analysis been performed appropriately and rigorously? 

Reviewer #2: N/A

Reviewer #3: Yes

4. Have the authors made all data underlying the findings in their manuscript fully available?

Reviewer #2: No

Reviewer #3: Yes

5. Is the manuscript presented in an intelligible fashion and written in standard English?

Reviewer #2: Yes

Reviewer #3: Yes

6. Review Comments to the Author

Reviewer #2: Paper is improved.

However, proofread well for the clarity and language improvements.

Since this paper is going to publish in 2025 , it would be better to include some more latest literature.

Reviewer #3: #1. Line no. 154 Hyperlink to mention the exact source and full path.

#2. Line Nos. 181 & 182 Mentioned "execution that does not need any installations," required import package installations or not for execution. Rephrase it.

#3. Line No. 186. GFLOPs for the first time mentioning the paragraph extent to full-length abbreviation.

#4. Line no. 210 and no. 210, no. 210, no. 210, and no. 210, no. 210, no.210,211 mentioned (1), (2), and (3) about table numbering or metric equation numbering not being addressed/referred to inside the paragraph text before or after the numbering with statement explanation.

#5. Line no. 221 after metric mentioning for precision and recall denominator "for precision, False Positives (FP)" and recall "False Negatives (FN):" Check the mentioned information and update accordingly.

#6. Equation no. 2: "gt" and "pr" full-length text abbreviation required.

#7. Line no . 243 in Equation no. 7 right form was "F1 Score = 2 ×" Precision × Recall / Precision + Recall," but it mentioned "Precision - Recall."

#8. Overall, all the equation numbering describes it before or after the numbering inside the paragraph with the reference number of the equation.

#9. Line no. 264 in the equation number 9 description area mentions "N is the total number of models," which is not present in equation no. 9.

#10. Figure 1 is required to be clear; the image is unclear and blurry.

#11. Figure 2.b labelled image caption titles are not clear.

#12. Figure 5 caption titles—type it clearly

* For your information : Note few mentioned equation numbering based on original submission.

7. PLOS authors have the option to publish the peer review history of their article (what does this mean? ). If published, this will include your full peer review and any attached files.

**Do you want your identity to be public for this peer review?** For information about this choice, including consent withdrawal, please see our Privacy Policy .

Reviewer #2: No

Reviewer #3: **Yes: ** Dr Janarthanan Sekar

---

## [Author Response · Author response to Decision Letter 2]

30 May 2025

Dear Academic Editor and Reviewers,

We sincerely appreciate your contribution. After carefully revising, we concurred with your comments and made the necessary revisions to ensure that our manuscript is in a better version and satisfies PLOS ONE's publishing requirements.

In "Response to Reviewers" file, we would like to respond to every concern highlighted by the academic editor and two reviewers throughout the review process, including:

Part 1. Responses to Academic Editor;

Part 2. Responses to Reviewer #2;

Part 3. Responses to Reviewer #3.

Best regards,

Tran Dang Hung

---

## [Editor Report · Decision Letter 2]

Mangrove species classification using a proposed ensemble U-Net model and Planet satellite imagery: A case study in Ngoc Hien district, Ca Mau province, Vietnam

PONE-D-24-60427R2

Dear Dr. Pham,

We’re pleased to inform you that your manuscript has been judged scientifically suitable for publication and will be formally accepted for publication once it meets all outstanding technical requirements.

Kind regards,

Rodrigo Nogueira Vasconcelos, Ph.D.

Academic Editor

PLOS ONE
---

## [Editor Report · Acceptance letter]

PONE-D-24-60427R2

PLOS ONE

Dear Dr. Pham,

I'm pleased to inform you that your manuscript has been deemed suitable for publication in PLOS ONE. Congratulations! Your manuscript is now being handed over to our production team.

Kind regards,

on behalf of

Dr. Rodrigo Nogueira Vasconcelos

Academic Editor

PLOS ONE